# Enhanced random forest with geologically-informed feature optimization for complex volcanic rock lithology identification: A case study in the Wangfu Fault Depression, Songliao Basin

**Xiu Jin[1], Taiji Yu[2,3]\*, Pujun Wang[3], Minglong Nie[2], Shichang Lu[1]**

**1** School of Business Administration, Liaoning Technical University, Huludao, Liaoning, China, **2** College of Safety Science and Engineering, Liaoning Technical University, Huludao, Liaoning, China, **3** College of Earth Sciences, Jilin University, Changchun, Jilin, China

\* yutj1988@126.com

## Abstract

Identifying lithologies within the volcanic reservoirs of the Huoshiling Formation (Wangfu Fault Depression, Songliao Basin) remains challenging due to extreme heterogeneity, limited core control, and ambiguous responses on conventional logs. We introduce an enhanced machine-learning framework for high-precision classification of these complex volcanic sequences, leveraging detailed core descriptions and five conventional well logs—gamma ray (GR), compensated neutron (CNL), bulk density (DEN), acoustic travel-time/sonic (AC), and deep array laterolog resistivity (RLA5)—from 12,388 depth-matched samples across 20 wells, encompassing 18 lithologies. The core innovation is an enhanced Random Forest (eRF) specifically engineered for geological and data-centric challenges. The eRF synergistically integrates: (1) Borderline-SMOTE to counteract severe class imbalance by selectively augmenting minority instances near decision boundaries, critical for rare but geologically significant facies; (2) C4.5 decision trees with gain-ratio splitting to optimize node-level feature selection from correlated continuous logs; and (3) Kendall's coefficient of concordance (Kendall's W) to stabilize feature-importance ranking across trees, prioritizing robust predictors. Against standard RF, back-propagation neural network (BPNN), k-nearest neighbors (kNN), and support-vector machine (SVM), the eRF attains 96.34% overall accuracy. Per-class accuracies exceed 0.88 for all 18 lithologies, with the largest improvement (+43 percentage points) for trachytic tuff. Sensitivity analysis indicates GR and AC dominate, together accounting for >60% of model decisions. This geologically attuned, optimized ensemble provides a transferable route to high-resolution lithology logs in uncored intervals, substantially aiding hydrocarbon sweet-spot prediction in complex volcanic settings.

**Data availability statement:** All data underlying the findings of this study are publicly available within the article. The source code for the enhanced Random Forest (eRF) model developed for this study is publicly available on GitHub. The repository can be accessed at: https://github.com/yuzc18/erf-volcanic-lithology. A DOI has been assigned to the repository and can be cited as follows: [10.5281/zenodo.17121940]. For additional data requests, researchers may also contact the Research Office of the School of Safety Science and Engineering, Liaoning Technical University (email: grants_ky@lntu.edu.cn).

**Funding:** This research was supported by the Liaoning Provincial Department of Education under Grant No. JYTQN2023207 (received by T.Y.) and the National Natural Science Foundation of China under Grant No. 41790453 (received by P.W.). The funders had no role in study design, data collection and analysis, decision to publish, or preparation of the manuscript.

**Competing interests:** The authors have declared that no competing interests exist.

## Introduction

The escalating global demand for hydrocarbons necessitates exploration in increasingly complex geological settings, with deep volcanic sequences emerging as critical targets [1]. Volcanic reservoirs are globally significant, documented in numerous basins worldwide; in China, high-yield discoveries in the Liaohe, Jilin, and Shengli oilfields underscore their potential [2]. The Songliao Basin—Northeast China's largest Late Mesozoic continental hydrocarbon-bearing basin—hosts substantial Cretaceous volcanic–sedimentary successions owing to its complex tectonic history [3–6]. Within this basin, the Huoshiling Formation of the Wangfu Fault Depression is a key exploration focus with significant natural-gas reserves [7] (Fig 1). To contextualize the classification task, a generalized lithostratigraphic section for the Huoshiling succession is shown in Fig 1b [8].

Accurate lithology identification in the Huoshiling Formation is hampered by the extreme complexity of its volcanic rock assemblages. The succession comprises 18 distinct lithologies—including lavas, pyroclastic lavas, pyroclastic rocks, and sedimentary volcaniclastic rocks—that frequently exhibit overlapping responses on conventional logs: gamma ray (GR), compensated neutron (CNL), bulk density (DEN), acoustic travel-time/sonic (AC), and deep array laterolog resistivity (RLA5). Rapid facies transitions, brecciation/vesiculation, and post-emplacement alteration (e.g., zeolitization, clay enrichment) further blur contrasts across gradational contacts, while limited core control for several lithologies constrains supervision and hampers validation [9–11]. Under these conditions, thin tuffaceous or brecciated interbeds embedded within thick lava packages are readily obscured in routine cross-plots and rule-based interpretations, yielding inconsistent and operator-dependent results [12].

Machine learning (ML) offers a scalable alternative by discovering implicit, nonlinear patterns in high-dimensional well-log data. Classical algorithms—neural networks, support-vector machines (SVMs), and decision-tree ensembles—have been applied to lithology prediction with encouraging outcomes in multiple basins [13,14]. Recent contributions further demonstrate both the promise and the methodological caveats: ML-based lithology prediction from conventional logs in the Cambay Basin with multi-classifier evaluation [15]; vertical lithological proxies using statistical and AI approaches in the Krishna–Godavari offshore [16]; a state-of-the-art synthesis on petrographic classification from geophysical logs reviewing data, features, and algorithm choices [12]; and ML-assisted petrographic classification in coal-measure successions from the Bokaro coalfield [17]. Collectively, these works highlight the potential of data-driven classification while emphasizing the need to address class imbalance, correlated predictors, and validation protocols aligned with geological structure and data provenance.

To address these challenges in the Huoshiling succession, we develop an enhanced Random Forest (eRF) tailored to volcanic-lithology identification. The approach couples geological realities with algorithmic mechanisms in a unified framework: Borderline-SMOTE concentrates synthetic augmentation at difficult decision boundaries to improve separability for minority lithologies; C4.5-style gain-ratio

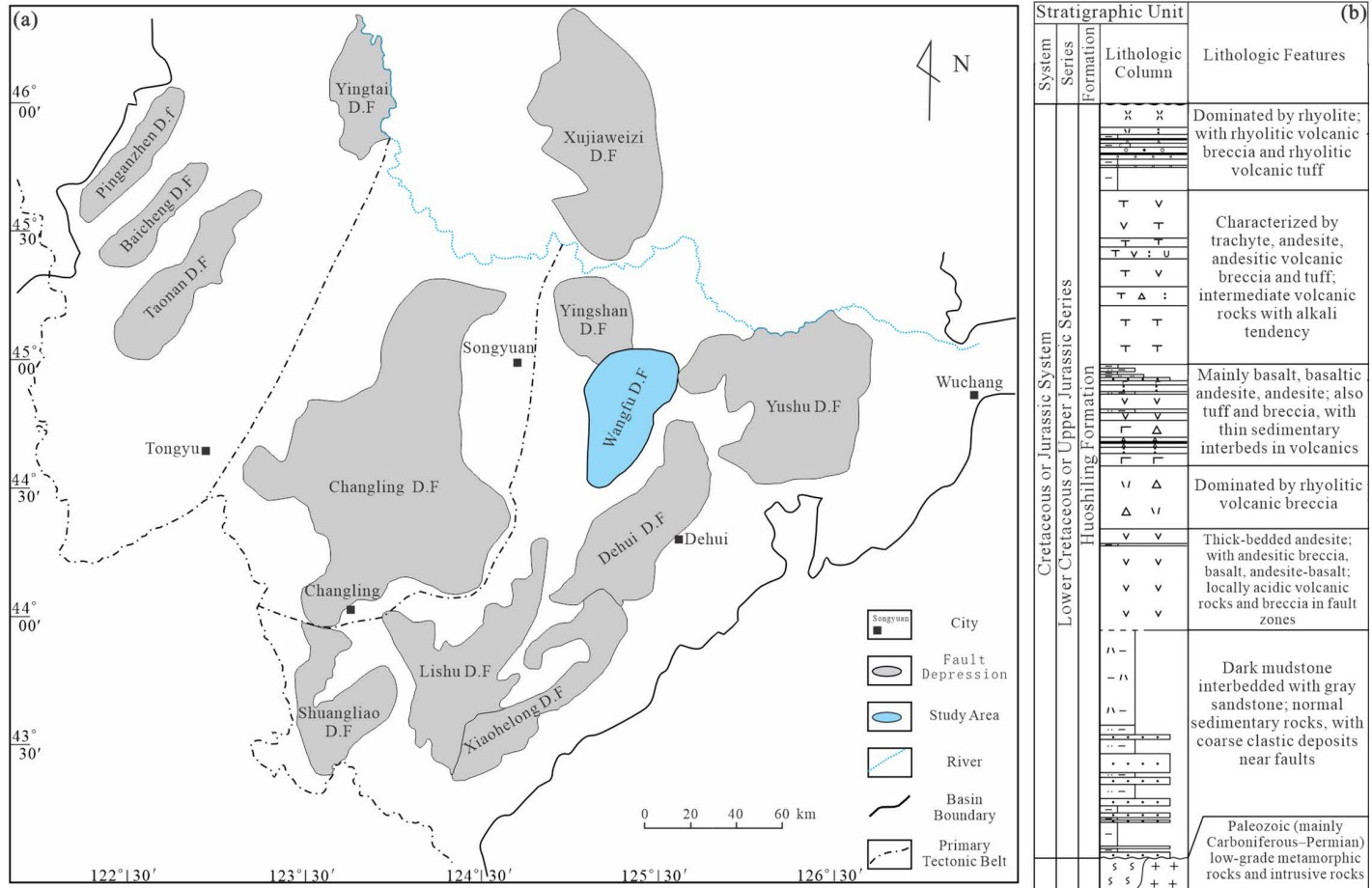

**Fig 1. Location and tectonic subdivision of the Songliao Basin.** (a) Distribution of fault depressions within the basin, highlighting the Wangfu Fault Depression; (b) Stratigraphic column of Huoshiling Formation succession.

splitting prioritizes geologically discriminative thresholds and mitigates the multi-valued-attribute bias of Gini when selecting among correlated continuous predictors; and Kendall's coefficient of concordance (Kendall's W) promotes stability in feature importance across trees, discouraging idiosyncratic splits and enhancing ensemble robustness. We evaluate the eRF against established baselines—standard RF, back-propagation neural network (BPNN), k-nearest neighbors (kNN), and SVM—using a multi-well dataset and rigorous model assessment. The generalized lithostratigraphic context (Fig 1b) anchors the classification task in the volcanic architecture of the Huoshiling succession and facilitates geological interpretation of the results.

## Geological setting and data sources

The Songliao Basin in northeastern China is the world's largest Late Mesozoic continental hydrocarbon-bearing basin. The Wangfu Fault Depression, in its southern sector, covers ~3,500 km² and hosts extensive volcanic successions within the Cretaceous Huoshiling and Shahezi formations. The Huoshiling Formation unconformably overlies Jurassic strata and is lithologically subdivided, from base to top, into trachyte, pyroclastic rocks, and rhyolite [18]. Data from 20 wells indicate a total thickness of volcanic rocks in the Huoshiling Formation reaching 9,679.69 m.

Following the volcanic-reservoir classification of Wang Pujun et al. [19], and integrating core observations with thin-section analyses, the Huoshiling succession in the study area comprises four structural classes and 18 principal lithologies (Fig 2; Table 1): volcanic lava, volcanic breccia, pyroclastic lava, and pyroclastic rock. These encompass diverse compositional types (e.g., trachyte, rhyolite, basalt) and multiple tuff and breccia variants, each with distinctive textures and fabrics that influence well-log responses. The large number of classes and their uneven representation (Table 1) underscore a pronounced class-imbalance challenge.

Core descriptions and five conventional well logs—gamma ray (GR), compensated neutron (CNL), bulk density (DEN), acoustic travel-time/sonic (AC), and deep array laterolog resistivity (RLA5)—were compiled from 20 wells as input features. Ground-truth lithology labels were derived from detailed core/cuttings analyses. In total, 12,388 depth-matched samples were assembled, each comprising a lithology label and five log readings. Prior to modeling, the logging data underwent quality control and normalization to correct depth mismatches and scale differences, yielding a consistent dataset for training and validation.

## Materials and methods

To achieve robust lithology identification, we compare the proposed enhanced Random Forest (eRF) against four widely used machine-learning algorithms: standard Random Forest (RF), back-propagation neural network (BPNN), k-nearest neighbors (kNN), and support-vector machine (SVM). All models are configured for 18-class prediction using five conventional logs (GR, CNL, DEN, AC, RLA5) as inputs. Hyperparameters are tuned by nested cross-validation with groups defined at the well level (Table 4).

### Standard machine learning algorithms (Baselines)

Random Forest (RF). The standard RF uses 100 decision trees with bootstrap resampling; node splits are determined by the Gini index, considering a random subset of features at each split (max_features = sqrt of total features). See Table 4 for complete settings.

Back-propagation Neural Network (BPNN). A feed-forward network is configured with one hidden layer (10 neurons, ReLU activation) and a softmax output layer for multiclass classification. Optimization is performed with Adam (Table 4).

k-Nearest Neighbors (kNN). This instance-based learner [20] classifies a sample by the majority class among its k nearest neighbors (k = 5, Euclidean distance) in the normalized feature space (Table 4).

Support-Vector Machine (SVM). A multiclass SVM [21] with one-vs-one strategy and an RBF kernel is employed. Hyperparameters (C and $\gamma$) are selected by grid search (Table 4).

### Enhanced random forest (eRF)

To address the specific challenges of Huoshiling lithology identification—namely, severe class imbalance among 18 lithotypes and the need to exploit subtle, overlapping log responses—we propose an eRF that integrates three components synergistically (Fig 3): Borderline-SMOTE for targeted imbalance correction at decision boundaries [22]; a C4.5-style gain-ratio probe to prioritize geologically discriminative thresholds on continuous logs [23]; and Kendall's coefficient of concordance (Kendall's W) to promote stability in feature usage across trees [24]. The source code for the enhanced Random Forest (eRF) model used in this study is publicly available on GitHub. The repository can be accessed at: https://github.com/yuzc18/erf-volcanic-lithology. A DOI has been assigned to the repository and can be cited as follows: [10.5281/zenodo.17121940].

**Borderline-SMOTE for imbalance correction.** The dataset exhibits pronounced imbalance (e.g., 2,670 trachyte samples vs. 116 basalt samples; Table 3), which biases learners toward majority classes. We therefore apply Borderline-SMOTE [22] during preprocessing. Unlike global oversampling, Borderline-SMOTE selectively synthesizes minority instances near decision boundaries—precisely where misclassification risk is highest—thereby increasing minority

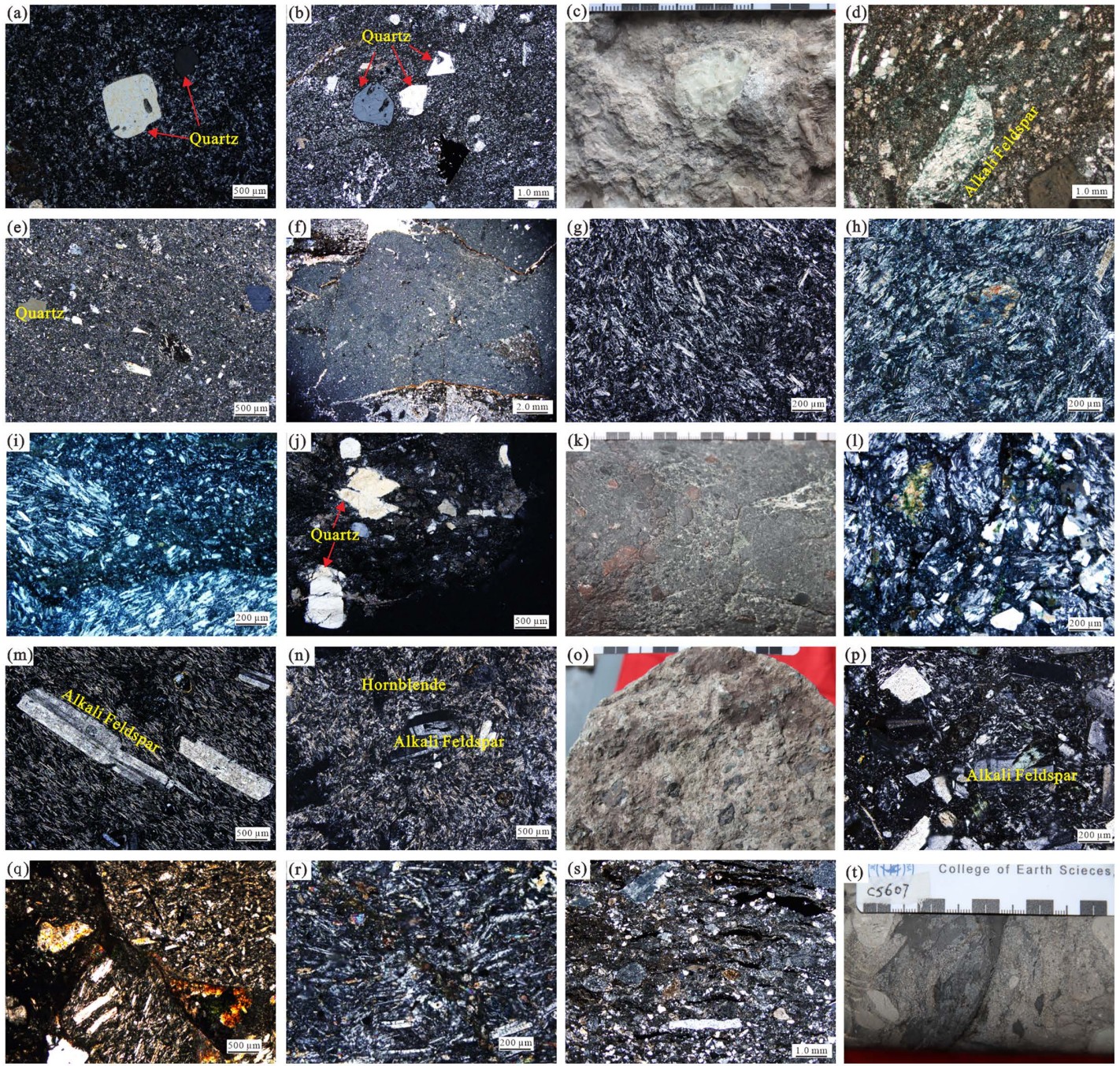

**Fig 2. Representative core photographs of volcanic lithologies from the Huoshiling Formation, Wangfu Fault Depression.** Each image corresponds to a specific volcanic rock type classified based on texture and composition. See captions (a–t) for sample depth and well ID. (a) Rhyolite, C9, 2270.12 m;(b) Rhyolitic tuff lava, C607, 2720.84 m;(c-d) Rhyolitic breccia lava, C14, 3005.85 m;(e) Rhyolitic tuff, C9, 2277.62 m;(f) Rhyolitic volcanic breccia, C14, 2292.82 m;(g) Andesite, C13, 2241.00 m;(h) Andesitic tuff lava, C11, 2769.00 m;(i) Andesitic breccia lava, C11, 2769.00 m;(j) Andesitic tuff, C4, 3929.00 m;(k-l) Andesitic volcanic breccia, C8, 2209.02 m;(m) Trachyte, C606, 2405.15 m;(n) Trachytic tuff lava, C6, 3020.00 m;(o) Trachytic breccia lava, C606, 2406.55 m;(p) Trachytic tuff, C14, 3090.00 m;(q) Trachytic volcanic breccia, C13, 2308.00 m;(r) Basalt, C10, 3324.00 m;(s) Sedimentary tuff, C9, 2279.52 m;(t) Sedimentary volcanic breccia, C607, 2740.04 m.

**Table 1. Lithology types and relative thicknesses of volcanic rocks in the Huoshiling Formation, Wangfu Fault Depression.**

| Structural Class | Compositional Class | Lithology Type (Relative Thickness) |
|---|---|---|
| Volcanic Lava (52.09%) | Basic (0.57%) | Basalt (0.57%) |
| | Intermediate (50.13%) | Trachyte (43.47%) |
| | | Andesite (6.66%) |
| | Acidic (1.39%) | Rhyolite (1.39%) |
| Pyroclastic Lava (17.27%) | Intermediate (15.34%) | Trachytic Brecciated Lava (11.02%) |
| | | Trachytic Tuff Lava (0.92%) |
| | | Andesitic Brecciated Lava (3.05%) |
| | | Andesitic Tuff Lava (0.35%) |
| | Acidic (1.93%) | Rhyolitic Brecciated Lava (1.00%) |
| | | Rhyolitic Tuff Lava (0.93%) |
| Pyroclastic Rocks (23.44%) | Intermediate (22.80%) | Trachytic Volcanic Breccia (16.07%) |
| | | Trachytic Tuff (3.43%) |
| | | Andesitic Volcanic Breccia (1.86%) |
| | | Andesitic Tuff (1.44%) |
| | Acidic (0.64%) | Rhyolitic Volcanic Breccia (0.14%) |
| | | Rhyolitic Tuff (0.50%) |
| Sedimentary Volcaniclastic Rocks (7.20%) | | Sedimentary Volcanic Breccia (1.11%) |
| | | Sedimentary Tuff (6.09%) |

Note: Percentages sum to 100% across all listed lithologies, reflecting their contribution to the total volcanic sequence studied.

density in ambiguous regions and improving recognition of rare but geologically significant lithologies. Given our data characteristics, this choice is preferable to alternatives such as NearMiss or ADASYN [25,26], which either remove informative boundary cases or may introduce noisier samples.

**Feature optimization with C4.5 decision trees.** To strengthen each tree within the ensemble, we adopt C4.5-style gain-ratio splitting [23]. The gain ratio normalizes information gain by split information, mitigating the tendency to favor attributes with many effective cut points—a known issue for the Gini index on continuous, potentially multi-valued predictors. By recursively selecting thresholds with the highest gain ratio, C4.5 emphasizes log features that are most discriminative for lithology and reduces interference from redundant parameters, yielding stronger base learners.

**Ensuring feature stability with Kendall's W.** To further refine feature utilization and enhance model stability, Kendall's coefficient of concordance (Kendall's W) [24] is integrated into the eRF framework. Kendall's W quantifies the consistency (agreement) of feature importance rankings derived from multiple decision trees within the ensemble. In this eRF, features demonstrating high concordance (i.e., consistently ranked as important across many trees) and strong correlation with lithology are prioritized or weighted more heavily during the construction of subsequent decision trees or in the feature selection process for node splitting. This mechanism aims to reduce information redundancy, enhance the diversity of effective features used by different trees, and ensure that the model relies on genuinely robust and stable discriminators rather than spurious correlations present in subsets of the data. This contributes to the overall stability and reliability of the ensemble's predictions.

By combining Borderline-SMOTE to rebalance informative boundaries, C4.5 gain-ratio to improve node-level feature selection, and Kendall's W to stabilize feature usage, the eRF is designed to learn geologically meaningful decision rules from noisy, imbalanced logs. The rationale for each enhancement module is summarized in Table 2.

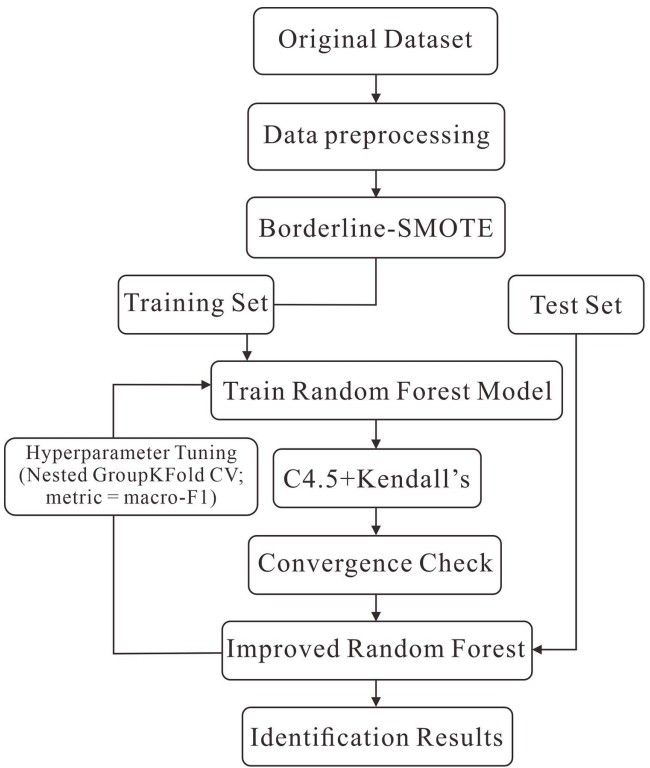

**Fig 3. Workflow of the proposed enhanced Random Forest (eRF) algorithm for volcanic lithology classification.** Data preprocessing includes quality control, Z-score standardization, and targeted imbalance correction with Borderline-SMOTE. The eRF is trained under nested GroupKFold cross-validation (inner loop: hyperparameter tuning by macro-F1; outer loop: model assessment). Trees adopt C4.5 gain-ratio splitting, and Kendall's W is used to promote stability in feature usage across trees. After aggregating outer-fold results, the final eRF is refit on the full training set and applied to the held-out test set and to blind wells in a zero-shot manner.

In summary, the eRF directly addresses the geological complexities of the study area by increasing sensitivity to minority lithologies and stabilizing feature usage via gain-ratio splits and Kendall's W, thereby improving robustness and generalization.

## Results

### Data preprocessing and model training

Data Preparation: A dataset of 12,388 samples from 18 volcanic lithologies was compiled from cored intervals of 20 wells in the Wangfu Fault Depression. Each sample (0.125 m interval) includes five logging parameters (GR, CNL, DEN, AC, RLA5) and a lithology label. Table 3 details sample counts and logging characteristics, highlighting severe class imbalance (e.g., trachyte: 2,670 samples; basalt: 116 samples) and overlapping log responses, which pose classification challenges. Despite overlaps, some lithologies show distinct signatures (e.g., basalt's low GR; rhyolitic tuff's high GR).

Data Standardization: Z-score normalization was applied to the five log curves to eliminate scale disparities:

$$x' = \frac{x - \mu}{\sigma}$$

(1)

where x is the raw logging value, μ is the sample mean, σ is the standard deviation, and x' is the normalised value.

**Table 2. Rationale for enhanced random forest (eRF) enhancement modules.**

| Enhancement Component | Specific Geological/Data Challenge Addressed | Brief Explanation of How the Component Addresses the Challenge | Example Key Parameters/Settings |
|---|---|---|---|
| Borderline-SMOTE | Severe class imbalance of critical minority lithologies; ambiguous log signatures at gradational or complex inter-lithotype boundaries. | Selectively generates synthetic samples around minority class instances near decision boundaries. This enhances the model's ability to learn in these ambiguous zones, reducing misclassification due to sparse data and improving recognition of rare but geologically significant lithologies. | k_neighbors for SMOTE, m_neighbors for identifying borderline points, oversampling ratio adjusted based on dataset imbalance. |
| C4.5 Decision Trees (using Gain Ratio) integrated into RF | Selection of the most discriminative logging features from noisy/correlated data for node splitting; potential bias of Gini index with multi-valued attributes in well log data. | C4.5 uses gain ratio as the splitting criterion, which is less biased towards features with many values and can be more robust for handling the varied nature of well log responses. This helps in selecting features more sensitive to lithological differences, improving individual tree efficacy. | Splitting criterion set to 'gain ratio'. |
| Kendall's W integrated into RF construction | Ensuring stability of feature importance in diverse volcanic facies; reducing model overfitting risk by relying on consistently important features. | Quantifies consistency of feature importance rankings across trees. Actively used to guide decision trees by assigning stability weights to features, modulating the C4.5 gain ratio. Prioritizes features that are both discriminative and consistently important, enhancing overall model robustness and reliability. | Integration method involves calculating Kendall's W iteratively or on subsets of trees to derive stability weights that influence feature selection in subsequent tree building. |
| Nested GroupKFold CV (by well) for hyperparameter tuning | Preventing well-level leakage and optimistic selection bias; robust model selection across spatial domains | Outer GroupKFold (by well) for model assessment; inner CV for hyperparameter search; macro-F1 as the selection metric; final refit on full training set | Group = well ID; K_outer, K_inner; selection = macro-F1; RF grid (n_estimators, max_depth, max_features, min_samples_leaf, bootstrap, class_weight); tuning/standardization fit on training folds only |

Following the protocol, data were split into a held-out test set (20%) and a training set (80%). Model selection on the training set used nested GroupKFold cross-validation (group = well): the inner loop performed grid search with macro-F1 as the selection metric, and the outer loop estimated generalization. Selected models were then refit on the full training set and evaluated on the held-out test set. Table 4 summarizes the deployed hyper-parameters; the Notes list the inner-CV search spaces. The decision threshold was 0.5 unless stated otherwise.

Test-set performance was reported by accuracy, precision, recall, and F1-score, following standard definitions for classification evaluation [27]. For the multiclass case, results are macro-averaged unless otherwise noted. For a given class c in a one-vs-rest setting with true positives TP, true negatives TN, false positives FP, and false negatives FN, the metrics are

Accuracy:

$$Accuracy = \frac{TP + TN}{TP + TN + FP + FN} \tag{2}$$

Precision:

$$Precision = \frac{TP}{TP + FP} \tag{3}$$

Recall:

**Table 3. Sample counts and statistical ranges of log parameters for volcanic lithologies.**

| Lithology/Sample Count | Statistic | GR (API) | CNL (%) | DEN (g·cm⁻³) | AC (µs·m⁻¹) | RLA5 (Ω·m) |
|---|---|---|---|---|---|---|
| Basalt/116 | Range/ Mean | 24.09–46.08 30.80 | 1.44–7.53 3.94 | 2.50–2.90 2.79 | 140.09–197.12 168.15 | 71.22–470.47 232.89 |
| Trachyte/2670 | Range/ Mean | 48.32–106.34 83.41 | 2.88–18.38 7.95 | 2.31–2.69 2.51 | 159.40–249.65 191.58 | 26.52–1999.26 546.91 |
| Andesite/361 | Range/ Mean | 37.78–86.24 61.08 | 3.92–15.21 7.70 | 2.35–2.75 2.61 | 158.35–240.06 191.19 | 30.34–3155.20 549.84 |
| Rhyolite/367 | Range/ Mean | 82.40–195.51 154.97 | 1.55–27.76 6.86 | 2.36–2.65 2.52 | 170.52–261.91 198.76 | 24.57–554.94 154.03 |
| Trachytic Brecciated Lava/1603 | Range/ Mean | 32.29–125.41 75.45 | 4.15–33.40 12.69 | 2.37–2.68 2.49 | 170.93–243.24 197.85 | 11.15–942.47 227.64 |
| Trachytic Tuff Lava/217 | Range/ Mean | 43.61–102.25 67.19 | 5.56–26.05 14.50 | 2.48–2.65 2.57 | 187.63–244.87 217.84 | 12.14–176.27 52.20 |
| Trachytic Volcanic Breccia/2046 | Range/ Mean | 35.18–157.00 67.90 | 5.60–17.42 10.08 | 1.98–2.63 2.51 | 168.25–241.07 191.01 | 19.15–926.36 266.79 |
| Andesitic Brecciated Lava/303 | Range/ Mean | 31.94–148.32 83.37 | 4.59–33.23 11.14 | 2.03–2.79 2.58 | 165.26–269.06 203.33 | 11.70–348.10 104.20 |
| Trachytic Tuff/704 | Range/ Mean | 32.60–100.76 56.30 | 3.38–30.77 10.31 | 2.15–2.74 2.60 | 145.08–239.11 194.69 | 24.61–998.18 213.64 |
| Andesitic Tuff Lava/121 | Range/ Mean | 33.34–83.91 58.18 | 4.97–27.71 16.45 | 2.37–2.64 2.49 | 170.12–253.87 202.53 | 31.83–348.16 157.25 |
| Andesitic Volcanic Breccia/839 | Range/ Mean | 24.51–140.06 55.79 | 3.39–22.98 13.95 | 2.12–2.74 2.49 | 170.56–257.26 222.42 | 10.74–3119.46 144.70 |
| Andesitic Tuff/174 | Range/ Mean | 45.56–86.83 60.82 | 13.89–20.42 16.95 | 2.35–2.61 2.50 | 195.78–248.20 227.02 | 13.02–29.80 18.97 |
| Rhyolitic Brecciated Lava/314 | Range/ Mean | 75.19–167.34 122.01 | 3.10–21.26 10.47 | 2.25–2.66 2.44 | 184.01–240.86 211.59 | 16.36–376.03 76.76 |
| Rhyolitic Tuff Lava/505 | Range/ Mean | 85.83–248.62 154.69 | 3.04–20.96 9.02 | 2.31–2.60 2.47 | 159.08–291.93 226.03 | 5.16–307.27 44.44 |
| Rhyolitic Volcanic Breccia/150 | Range/ Mean | 82.40–243.80 176.20 | 3.39–29.85 9.65 | 1.74–2.67 2.50 | 168.30–268.89 204.98 | 30.28–248.82 86.92 |
| Rhyolitic Tuff/154 | Range/ Mean | 118.11–282.62 181.90 | 6.82–35.18 14.76 | 2.11–2.69 2.43 | 201.94–318.89 239.72 | 4.38–129.81 28.41 |
| Sedimentary Volcanic Breccia/1237 | Range/ Mean | 43.37–149.40 103.98 | 5.03–20.68 12.79 | 2.33–2.61 2.49 | 182.77–271.47 213.11 | 18.51–270.88 114.16 |
| Sedimentary Tuff/509 | Range/ Mean | 49.49–259.91 135.16 | 4.06–29.73 14.61 | 1.78–2.71 2.50 | 185.41–298.71 219.31 | 6.74–99.49 45.69 |

$$Recall = \frac{TP}{TP + FN} \qquad (4)$$

F1-score:

$$F1\text{--}score = 2 \times \frac{Precision \times Recall}{Precision + Recall} \qquad (5)$$

Where TP = True Positives, TN = True Negatives, FP = False Positives, FN = False Negatives, generalized for multi-class via averaging.

**Table 4. Key hyperparameters and training settings for each machine-learning model.**

| Model | Key hyperparameters and settings |
|---|---|
| Enhanced RF (Proposed) | Number of decision trees = 300; Sampling method = Borderline-SMOTE; Splitting criterion = C4.5 gain ratio; Feature consistency screening = Kendall's concordance coefficient |
| Random Forest | Number of decision trees = 100; Splitting criterion = Gini index; Maximum features = sqrt(feature dimensions) |
| BPNN | Hidden layers = 1 layer with 10 neurons; Learning rate = 0.01; Activation function = ReLU; Training epochs = 500 (early stopping) |
| kNN | Neighbors (K) = 5; Distance metric = Euclidean |
| SVM | Kernel function = RBF (Radial Basis Function); Penalty parameter (C) = 10; Kernel coefficient (γ) = scale |

Notes. Inner CV search spaces—eRF: n_estimators in {100,200,300}; max_features in {sqrt, log2}; n_bins in {8,16,32}; Borderline-SMOTE: k_neighbors in {3,5,7}, m_neighbors in {8,10,12}, kind in {borderline-1, borderline-2}; stability subset: MIN_TOPK in {6,10}, MAX_TOPK in {16,20}. RF: n_estimators in {100,200,300}; max_features in {sqrt, log2}; min_samples_leaf in {1,2,4}. BPNN: hidden units in {8,10,16}; learning rate in {1e-2, 5e-3, 1e-3}; batch size in {32,64}; dropout in {0,0.2}. kNN: k in {3,5,7,9}; metric in {Euclidean, Manhattan}; weights in {uniform, distance}. SVM (RBF): C in {1,10,100}; gamma in {scale, 0.1, 0.01}. Protocol: nested GroupKFold (group = well), selection metric = macro-F1; decision threshold = 0.5.

## Lithology identification results

Table 5 summarizes the overall performance metrics. The proposed eRF model achieved the highest accuracy (96.34%) and F1-score (0.9635), significantly outperforming standard RF (87.73% accuracy), BPNN (86.97%), kNN (94.84%), and SVM (82.32%). The balanced precision and recall for eRF indicate robust classification across all lithologies.

Per-class accuracies (Table 6, Fig 4) further highlight the eRF's superiority. The eRF achieved >0.88 accuracy for all 18 lithologies. Notably, for trachytic tuff (a minority class), accuracy improved from 0.5455 (standard RF) to 0.9787 (eRF). Other models showed weaknesses: standard RF struggled with minority classes; BPNN underperformed for classes like sedimentary tuff (0.7087 vs. eRF's 0.9771); kNN, while competitive for some distinct lithologies, showed performance degradation for those with overlapping features; SVM had the lowest overall per-class performance.

Fig 4 shows that eRF delivers the sharpest diagonal and the sparsest off-diagonals. Residual errors concentrate within compositionally/texturally related pairs where log responses overlap—e.g., Trachytic Tuff Lava vs Trachytic Tuff, Trachytic Brecciated Lava vs Trachytic Volcanic Breccia, Andesitic Tuff Lava vs Andesitic Tuff, and their breccia counterparts; limited cross-confusion also appears inside the rhyolitic suite. Compared with the standard RF, eRF markedly suppresses leakage from minority pyroclastic units into dominant lavas, especially for Trachytic Tuff, consistent with Table 6. Baselines show characteristic weaknesses: BPNN exhibits broad low-amplitude bands under imbalance; kNN is competitive for distinctive end-members but degrades for overlapping pairs; SVM yields the weakest diagonal. These matrices corroborate that eRF improves not only aggregate metrics but also error structure.

Fig 5 (panels a–r) shows that the enhanced Random Forest (eRF) discriminates most lithologies very well: the one-vs-rest ROC curves cluster near the upper-left, with uniformly high AUCs (macro-AUC ≈ 0.999; most AUCs ≥ 0.98). At

**Table 5. Overall evaluation metrics for volcanic lithology identification using different machine-learning models.**

| Model | Accuracy | Precision | Recall | F1-Score |
|---|---|---|---|---|
| Enhanced RF (Proposed) | 0.9634 | 0.9635 | 0.9636 | 0.9635 |
| Random Forest | 0.8773 | 0.8733 | 0.8568 | 0.8564 |
| BPNN | 0.8697 | 0.8703 | 0.8698 | 0.8687 |
| kNN | 0.9484 | 0.9485 | 0.9488 | 0.9484 |
| SVM | 0.8232 | 0.8249 | 0.8242 | 0.8227 |

**Table 6. Per-class identification accuracy for all 18 volcanic lithologies across five classification models.**

| Lithology | Enhanced RF | Random Forest | BPNN | kNN | SVM |
|---|---|---|---|---|---|
| Basalt | 0.9961 | 1.0000 | 1.0000 | 1.0000 | 1.0000 |
| Trachyte | 0.9099 | 0.8944 | 0.8087 | 0.8949 | 0.7899 |
| Trachytic Brecciated Lava | 0.8877 | 0.8593 | 0.7332 | 0.8238 | 0.6753 |
| Trachytic Tuff Lava | 0.9939 | 0.8125 | 0.9476 | 0.9878 | 0.9002 |
| Trachytic Volcanic Breccia | 0.9335 | 0.9289 | 0.8080 | 0.8891 | 0.7560 |
| **Trachytic Tuff** | 0.9787 | 0.5455 | 0.7838 | 0.9627 | 0.6465 |
| Andesite | 0.9226 | 0.6885 | 0.8110 | 0.9075 | 0.7509 |
| Andesitic Brecciated Lava | 0.8982 | 0.7667 | 0.7545 | 0.9122 | 0.6906 |
| Andesitic Tuff Lava | 0.9890 | 0.7143 | 0.9753 | 0.9945 | 0.9452 |
| Andesitic Volcanic Breccia | 0.9856 | 0.9636 | 0.8531 | 0.9550 | 0.8216 |
| Andesitic Tuff | 1.0000 | 1.0000 | 1.0000 | 1.0000 | 1.0000 |
| Rhyolite | 0.9560 | 0.8831 | 0.8896 | 0.9744 | 0.8590 |
| Rhyolitic Brecciated Lava | 0.9944 | 0.9091 | 0.9487 | 0.9793 | 0.8853 |
| Rhyolitic Tuff Lava | 0.9881 | 0.9652 | 0.8812 | 0.9802 | 0.8399 |
| Rhyolitic Volcanic Breccia | 0.9708 | 0.8333 | 0.9189 | 0.9572 | 0.8872 |
| **Rhyolitic Tuff** | 0.9908 | 0.7419 | 0.9595 | 0.9889 | 0.9004 |
| Sedimentary Volcanic Breccia | 0.9722 | 0.9032 | 0.8748 | 0.9593 | 0.7907 |
| **Sedimentary Tuff** | 0.9771 | 0.8661 | 0.7087 | 0.9067 | 0.6114 |

the operating threshold of 0.5 (orange markers), false-positive rates remain near zero while true-positive rates span ~0.82–0.99. Slightly lower TPRs are observed for several transitional or texturally heterogeneous classes (b, c, e, g, h, o), consistent with overlapping log responses near facies contacts.

Panel (s) indicates small train–test gaps across classes, with no systematic drop on the test set, suggesting limited overfitting and stable generalization of the eRF. In conjunction with the method design, these observations align with the intended mechanism: Borderline-SMOTE enriches boundary cases in rare/ambiguous facies, C4.5 gain-ratio splits exploit subtle log sensitivities, and Kendall's W promotes stability in feature usage across trees. Together, these elements underpin the eRF's effectiveness for multi-class volcanic-lithology identification from conventional logs and motivate the subsequent discussion on geological factors and algorithmic synergy.

## Discussion

### Influence of geological factors on model design and success

The Huoshiling volcanic succession comprises 18 lithologies with diverse mineralogy and textures, and many contacts are gradational rather than sharp. Intra-facies textural variability, variable welding degree, and changes in clast–matrix proportions or alteration can make gamma ray (GR), compensated neutron (CNL), bulk density (DEN), acoustic travel-time/sonic (AC), and deep array laterolog resistivity (RLA5) respond in similar ways even when the underlying lithologies are distinct. These geological realities make class boundaries intrinsically fuzzy on conventional logs and motivate an ensemble-based solution that can accommodate multi-modal distributions and correlated continuous predictors. Accordingly, we selected Random Forest (RF) as the base learner and constructed an enhanced Random Forest (eRF) that embeds algorithmic elements tailored to the data-generating processes in this setting.

The five logs were chosen because they are physically sensitive to composition, porosity, fluid content, welding, and clast–matrix architecture, which together govern the petrophysical contrasts among lavas, pyroclastic lavas, pyroclastic rocks, and sedimentary volcaniclastic rocks. Within eRF, C4.5 gain-ratio splitting is used to identify informative thresholds

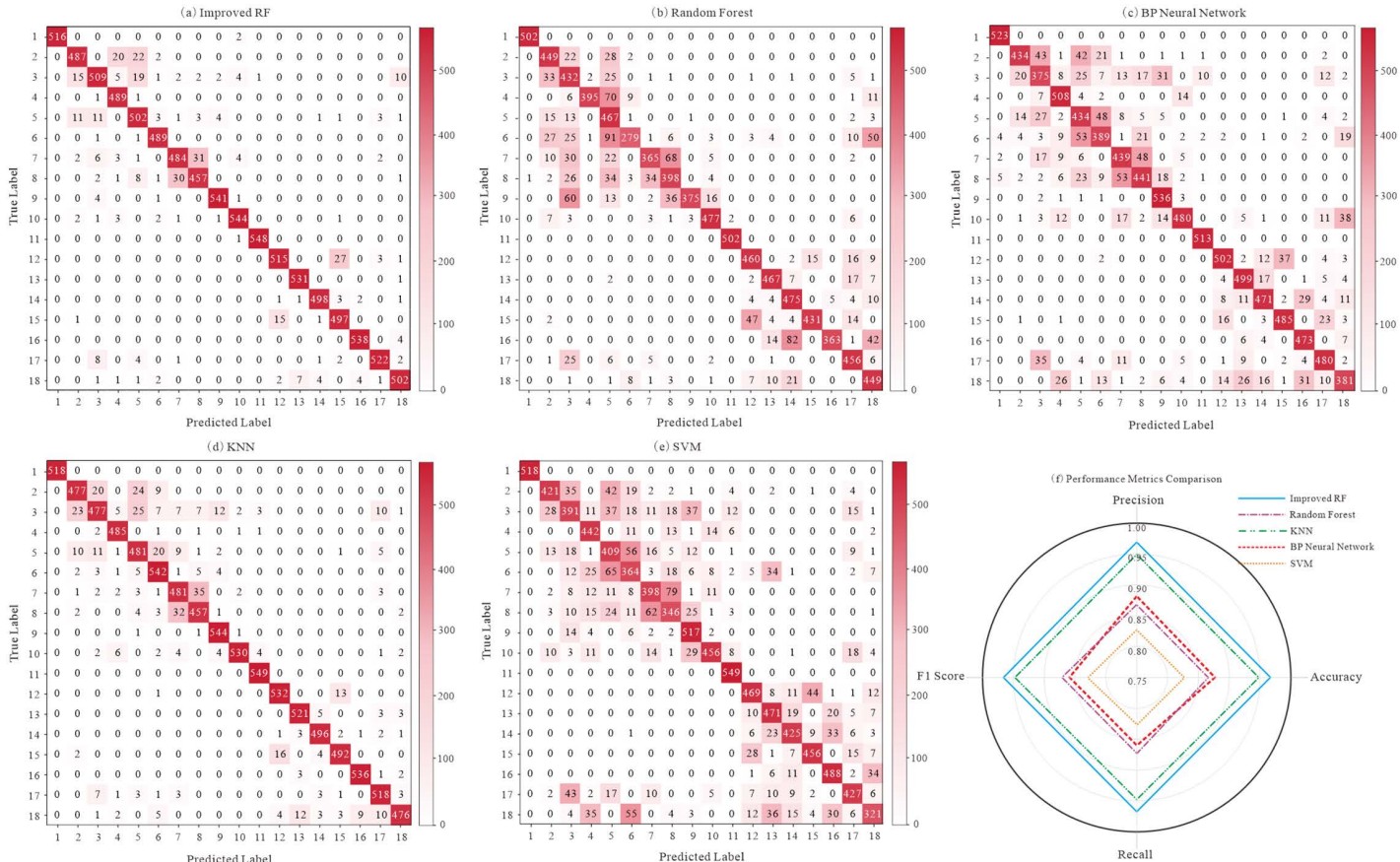

**Fig 4. Confusion matrices for lithology classification using five machine-learning models.** Each matrix shows the classification performance for 18 volcanic lithologies. The enhanced Random Forest (eRF) model exhibits the highest diagonal accuracy and lowest misclassification rates.(Lithology labels 1–18 correspond to: 1. Basalt; 2. Trachyte; 3. Trachytic brecciated lava; 4. Trachytic tuff lava; 5. Trachytic volcanic breccia; 6. Trachytic tuff; 7. Andesite; 8. Andesitic brecciated lava; 9. Andesitic tuff lava; 10. Andesitic volcanic breccia; 11. Andesitic tuff; 12. Rhyolite; 13. Rhyolitic brecciated lava; 14. Rhyolitic tuff lava; 15. Rhyolitic volcanic breccia; 16. Rhyolitic tuff; 17. Sedimentary volcanic breccia; 18. Sedimentary tuff).

on these continuous predictors while mitigating the bias of criteria that favor attributes with many effective cut points. Borderline-SMOTE is introduced because the scarcity of certain facies—thin tuffs, breccias, or transitional units—is a geological fact rather than an artifact of sampling, and the most consequential ambiguities occur precisely at inter-facies boundaries where minority instances are under-represented. Kendall's coefficient of concordance (Kendall's W) is incorporated to promote stability by favoring features whose importance is repeatedly high across trees, thereby limiting reliance on idiosyncratic splits that may be induced by local noise in complex volcanic sequences.

Viewed through this geological lens, the confusion matrices in Fig 4 are not arbitrary. The remaining off-diagonal entries concentrate within geologically related pairs whose petrophysical responses are known to overlap near facies contacts and within texturally heterogeneous intervals. Typical examples occur between Trachytic Tuff Lava and Trachytic Tuff; between Trachytic Brecciated Lava and Trachytic Volcanic Breccia; between Andesitic Tuff Lava and Andesitic Tuff; between Andesitic Brecciated Lava and Andesitic Volcanic Breccia; and within the rhyolitic suite between brecciated end-members and volcanic-breccia end-members. Occasional confusion is also observed between Sedimentary Volcanic Breccia and volcanic-breccia classes. In each of these cases, similar combinations of GR, DEN, CNL, AC, and RLA5 arise from comparable mineral assemblages and fabric, as well as from gradational contacts that attenuate sharp log contrasts.

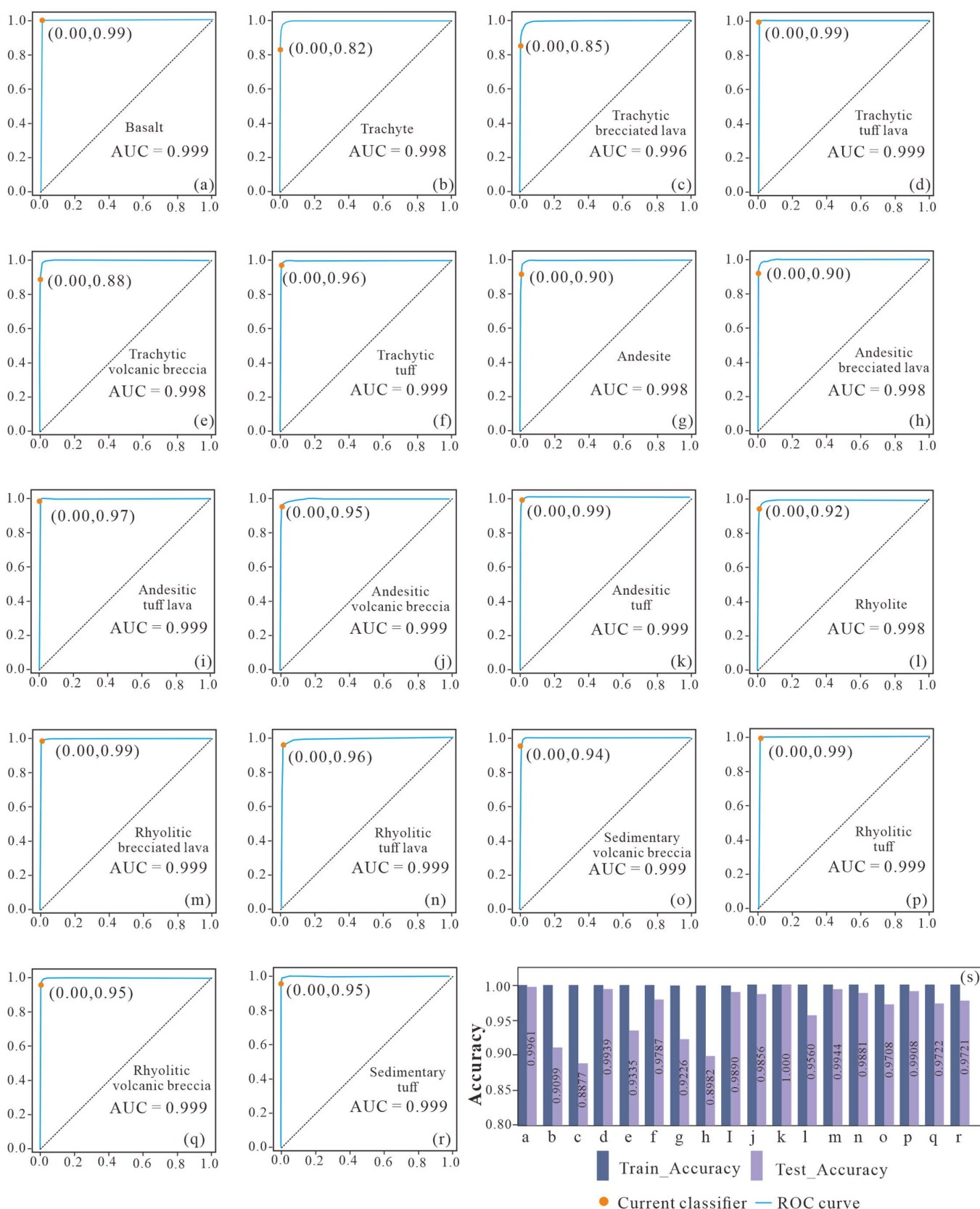

**Fig 5. Class-wise ROC on the test set for the enhanced Random Forest (eRF) (18 lithologies) with eRF train–test accuracy comparison.** Panels a–r display one-vs-rest ROC curves for each lithology; the orange dot marks the operating point at threshold = 0.5, and the parenthetical label gives its (FPR, TPR). The bottom box in each panel reports the class name and AUC (three decimals, truncated). Panel (s) shows per-class train/test one-vs-rest accuracy; x-axis letters a–r correspond to the ROC panels. Letter–lithology mapping: a Basalt; b Trachyte; c Trachytic brecciated lava; d Trachytic tuff lava; e Trachytic volcanic breccia; f Trachytic tuff; g Andesite; h Andesitic brecciated lava; i Andesitic tuff lava; j Andesitic volcanic breccia; k Andesitic tuff; l Rhyolite; m Rhyolitic brecciated lava; n Rhyolitic tuff lava; o Sedimentary volcanic breccia; p Rhyolitic tuff; q Rhyolitic volcanic breccia; r Sedimentary tuff.

The pattern is therefore geologically plausible and provides a diagnostic target for the algorithmic design choices embedded in eRF. Taken together, these geologically plausible confusions also expose where a vanilla RF is prone to fail in this setting: random feature selection may not consistently prioritize the most geologically informative log parameters at node splits; pronounced class imbalance tends to bias decisions toward dominant lithologies and suppress recall for rare but diagnostically important facies; and high structural similarity among trees can limit the ensemble's capacity to resolve multi-faceted boundaries. In the Huoshiling dataset, these effects are amplified by domain specifics—thin tuffaceous or brecciated interbeds embedded within thick lava packages, strongly correlated and multi-valued continuous predictors in conventional logs, and weak separability near gradational contacts—so boundary cases become both the most informative and the most easily misclassified. These considerations directly motivate the targeted enhancements embedded in the eRF (Borderline-SMOTE, C4.5 gain-ratio splitting, and Kendall's W), and they are consistent with the residual off-diagonal patterns in Fig 4 and the class-wise improvements for eRF in Fig 5 and Table 6.

## Effectiveness and synergy of algorithmic optimizations in eRF

The empirical improvements delivered by eRF can be traced to the coordinated roles of its three components and their alignment with the geological sources of ambiguity. Borderline-SMOTE enriches the local density of minority instances specifically at transition zones, which are the regions of greatest confusion in Fig 4. This increases the chance that decision boundaries are learned where they matter most rather than being pulled by the statistical dominance of thick lava units. C4.5 gain-ratio splitting then identifies thresholds on continuous logs that exploit subtle but physically meaningful contrasts while reducing the bias that would otherwise favor highly partitionable attributes. Kendall's W promotes ensemble consistency by prioritizing features that are repeatedly discriminative across trees, which curbs idiosyncratic splits that could widen scattered off-diagonals.

These mechanisms are reflected both in the aggregate metrics and in the structure of errors. Relative to the standard RF, eRF produces the sharpest diagonals and the sparsest off-diagonals in Fig 4, with especially clear suppression of leakage from minority pyroclastic units into dominant trachytic or andesitic lavas. The benefit is most striking for Trachytic Tuff, where accuracy rises from 0.5455 with standard RF to 0.9787 with eRF (Table 6). The baseline methods exhibit characteristic weaknesses that align with their inductive biases under severe class imbalance and overlapping clusters: back-propagation neural networks show broader low-amplitude off-diagonals; k-nearest neighbors remains competitive for distinctive end-members but degrades for overlapping trachytic and andesitic pairs; and support vector machines with radial kernels display the weakest diagonal when skew is pronounced. The eRF-only ROC panels in Fig 5 further corroborate these observations. The one-vs-rest curves cluster near the upper-left, macro-AUC is approximately 0.999 with most class AUCs at or above 0.98, and at the default operating threshold of 0.5 false-positive rates are near zero while true-positive rates span approximately 0.82 to 0.99 across classes. Panel (s) shows small train–test gaps, which is consistent with the stability imparted by Kendall's W and indicates that the observed gains are not the by-product of overfitting.

From a methodological perspective, the novelty here is not in any single component taken in isolation but in their integrated use within an RF framework that is explicitly tuned to the geological context of 18-class volcanic lithology identification from conventional logs. Addressing imbalance at critical boundaries, optimizing per-node splits for continuous and

potentially multi-valued predictors, and enforcing stability of feature usage across trees are complementary interventions. Their synergy explains why the error structure itself changes in Fig 4 in a way that is geologically meaningful and why the ROC behavior in Fig 5 is uniformly strong across facies, including many of the previously hard-to-separate minority types.

### External blind-well validation and generalization

Spatial generalization was assessed by zero-shot inference on an external blind-well set comprising 2,964 depth samples from five Huoshiling wells (C_1, C_8, C_9, C_10, C_21) that were entirely excluded from model development. The final eRF trained on the source field was applied directly—without retraining, without calibration, and without any resampling on blind data—using the same five logs as inputs. Ground-truth labels were compiled from the operator's well documentation, and intervals with missing or ambiguous labels were removed. Preprocessing mirrored the training pipeline with parameters frozen from the training stage: rows with any missing log value were discarded; the StandardScaler and label encoder fitted on the training set were reused to transform logs and map lithology codes; and predictions were generated once at the default threshold of 0.5.

Under this zero-shot setting, eRF achieved an average accuracy of 92.24%, with macro Precision, Recall, and F1 of 0.900, 0.906, and 0.903, respectively (Table 7). These values exceed those of the baselines—standard RF at 87.63%, k-nearest neighbors at 87.37%, back-propagation neural network at 81.38%, and support vector machine at 78.27%—by between 4.6 and 14.0 percentage points in accuracy. The per-class results corroborate robust transfer across lithologies: accuracies remain high for most classes, such as Basalt at 1.0000, Sedimentary Volcanic Breccia at 0.9670, and Trachytic Volcanic Breccia at 0.9360. Performance is relatively lower for transitional or texturally heterogeneous facies, such as Rhyolitic Volcanic Breccia at 0.7941 and Trachytic Tuff at 0.8429, which is consistent with gradational contacts

**Table 7. Lithology classification performance on blind test wells using different machine-learning models.**

| Lithology | Enhanced RF | Random Forest | BPNN | kNN | SVM |
|---|---|---|---|---|---|
| Basalt | 1.0000 | 1.0000 | 1.0000 | 1.0000 | 1.0000 |
| Trachyte | 0.9295 | 0.9057 | 0.8448 | 0.8817 | 0.8197 |
| Trachytic Brecciated Lava | 0.8927 | 0.8554 | 0.7487 | 0.8105 | 0.6859 |
| Trachytic Tuff Lava | 0.9074 | 0.8519 | 0.9074 | 0.9259 | 0.9074 |
| Trachytic Volcanic Breccia | 0.9360 | 0.9023 | 0.8060 | 0.8770 | 0.7840 |
| **Trachytic Tuff** | 0.8429 | 0.6842 | 0.7571 | 0.8684 | 0.6143 |
| Andesite | 0.8750 | 0.6778 | 0.8250 | 0.7444 | 0.8500 |
| Andesitic Brecciated Lava | 0.8889 | 0.8011 | 0.6914 | 0.7784 | 0.7037 |
| Andesitic Tuff Lava | 0.8571 | 0.6000 | 0.9286 | 0.9048 | 0.9286 |
| Andesitic Volcanic Breccia | 0.9754 | 0.9571 | 0.8768 | 0.9286 | 0.8571 |
| Andesitic Tuff | 0.9024 | 0.9535 | 0.9756 | 0.9070 | 0.9512 |
| Rhyolite | 0.9101 | 0.8696 | 0.7865 | 0.8696 | 0.7528 |
| Rhyolitic Brecciated Lava | 0.9054 | 0.8718 | 0.8919 | 0.9103 | 0.7973 |
| Rhyolitic Tuff Lava | 0.9435 | 0.9435 | 0.8226 | 0.9048 | 0.7823 |
| Rhyolitic Volcanic Breccia | 0.7941 | 0.7368 | 0.8235 | 0.8158 | 0.8235 |
| **Rhyolitic Tuff** | 0.8718 | 0.7692 | 0.9487 | 0.9487 | 0.8974 |
| Sedimentary Volcanic Breccia | 0.9670 | 0.9353 | 0.8548 | 0.9450 | 0.8152 |
| **Sedimentary Tuff** | 0.9060 | 0.8110 | 0.6154 | 0.8504 | 0.6068 |
| Average Accuracy | 0.9224 | 0.8763 | 0.8138 | 0.8737 | 0.7827 |
| Average Precision | 0.8998 | 0.8725 | 0.7572 | 0.8337 | 0.7193 |
| Average Recall | 0.9058 | 0.8400 | 0.8392 | 0.8833 | 0.8098 |
| Average F1-Score | 0.9025 | 0.8535 | 0.7834 | 0.8543 | 0.7465 |

and overlapping log responses. Taken together with the small train–test gaps observed earlier for eRF in Fig 5, panel (s), these blind-well outcomes support limited overfitting and stable generalization beyond the training area, and they indicate that the learned decision rules transfer across fields under spatial domain shift.

## Conclusions

This study developed and validated an enhanced Random Forest (eRF) for identifying 18 volcanic lithologies in the Wangfu Fault Depression (Huoshiling Formation, Songliao Basin) from five conventional well logs. The main conclusions are:

(i) The Huoshiling volcanic succession exhibits high lithological diversity (18 classes) and pronounced class imbalance; gradational contacts produce overlapping log signatures, which complicate automated identification. A labeled dataset integrating core/cuttings descriptions with GR, AC, DEN, CNL, and RLA5 was established accordingly.

(ii) The proposed eRF—synergistically integrating Borderline-SMOTE for targeted imbalance correction at decision boundaries, C4.5 decision trees (gain-ratio splitting) for optimized feature thresholds on continuous logs, and Kendall's coefficient of concordance to guide stable feature prioritization across trees—achieved a test-set accuracy of 96.34% with macro-F1 = 0.9635, indicating balanced performance across all classes.

(iii) eRF outperformed four classical machine-learning algorithms (standard RF, BPNN, kNN, SVM) on both overall and per-class metrics. The enhancements effectively address class imbalance at critical boundaries, improve split quality with a robust criterion, and promote stability of feature usage in the ensemble, yielding excellent recognition for all lithologies, especially previously hard-to-classify minority types and facies with ambiguous boundary signatures.

(iv) In external blind-well evaluation (zero-shot; five wells; 2,964 depth samples), eRF attained 92.24% average accuracy with macro Precision/Recall/F1 = 0.900/0.906/0.903, demonstrating robust spatial generalization beyond the training area and supporting deployment in similar multi-class volcanic settings to generate high-resolution lithological profiles in uncored intervals.

## Acknowledgments

We would like to thank working group of Volcanic Reservoirs and their Exploration, Jilin University, Changchun, China for their helps with field work. We also thank the two reviewers and Academic Editor Hu Li for the constructive reviews that significantly improved the manuscript.

## Author contributions

**Conceptualization:** Xiu Jin, Taiji Yu.

**Data curation:** Pujun Wang.

**Formal analysis:** Xiu Jin.

**Funding acquisition:** Taiji Yu,  Pujun Wang.

**Investigation:** Xiu Jin, Shichang Lu.

**Methodology:** Xiu Jin, Taiji Yu.

**Project administration:** Taiji Yu.

**Resources:** Taiji Yu, Pujun Wang.

**Software:** Xiu Jin, Minglong Nie.

**Supervision:** Taiji Yu, Pujun Wang.

**Validation:** Xiu Jin, Minglong Nie.

**Visualization:** Shichang Lu.

**Writing – original draft:** Xiu Jin.

**Writing – review & editing:** Xiu Jin, Taiji Yu.

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
