## [Decision Letter · Decision Letter 0]

8 Aug 2025

Dear Dr. Yu,

Thank you for submitting your manuscript to PLOS ONE. After careful consideration, we feel that it has merit but does not fully meet PLOS ONE’s publication criteria as it currently stands. Therefore, we invite you to submit a revised version of the manuscript that addresses the points raised during the review process.

We look forward to receiving your revised manuscript.

Kind regards,

Hu Li

Academic Editor

PLOS ONE

Journal Requirements:

[This research was supported by the Liaoning Provincial Department of Education under Grant No. JYTQN2023207 (received by T.Y.) and the National Natural Science Foundation of China under Grant No. 41790453 (received by T.Y.). The funders had no role in study design, data collection and analysis, decision to publish, or preparation of the manuscript.].

4. Thank you for stating the following in your manuscript:

[This work was supported by the Basic Scientific Research Projects of Liaoning Provincial Department of Education (grant no. JYTQN2023207) and the National Natural Science Foundation of China (grant no. 41790453). The funders had no role in study design, data collection and analysis, decision to publish, or preparation of the manuscript.]

[This research was supported by the Liaoning Provincial Department of Education under Grant No. JYTQN2023207 (received by T.Y.) and the National Natural Science Foundation of China under Grant No. 41790453 (received by T.Y.). The funders had no role in study design, data collection and analysis, decision to publish, or preparation of the manuscript.]

5. In the online submission form you indicate that your data is not available for proprietary reasons and have provided a contact point for accessing this data. Please note that your current contact point is a co-author on this manuscript. According to our Data Policy, the contact point must not be an author on the manuscript and must be an institutional contact, ideally not an individual. Please revise your data statement to a non-author institutional point of contact, such as a data access or ethics committee, and send this to us via return email. Please also include contact information for the third party organization, and please include the full citation of where the data can be found.

7. In the online submission form, you indicated that [All relevant data generated during this study are available within the manuscript and its Supporting Information files. The source code for the enhanced Random Forest model and the lithology classification results are available from the corresponding author upon reasonable request.].

8. We note that Figure 1 in your submission contain map images which may be copyrighted. All PLOS content is published under the Creative Commons Attribution License (CC BY 4.0), which means that the manuscript, images, and Supporting Information files will be freely available online, and any third party is permitted to access, download, copy, distribute, and use these materials in any way, even commercially, with proper attribution. For these reasons, we cannot publish previously copyrighted maps or satellite images created using proprietary data, such as Google software (Google Maps, Street View, and Earth). For more information, see our copyright guidelines: http://journals.plos.org/plosone/s/licenses-and-copyright.

Additional Editor Comments :

We sincerely appreciate your submission. Kindly revise the manuscript in accordance with the reviewers’ comments, paying particular attention to ensuring full compliance with the journal’s publication format.

Reviewers' comments:

Reviewer's Responses to Questions

**Comments to the Author**

1. Is the manuscript technically sound, and do the data support the conclusions?

Reviewer #1: Yes

Reviewer #2: Yes

2. Has the statistical analysis been performed appropriately and rigorously?

Reviewer #1: Yes

Reviewer #2: Yes

3. Have the authors made all data underlying the findings in their manuscript fully available?

Reviewer #1: Yes

Reviewer #2: Yes

4. Is the manuscript presented in an intelligible fashion and written in standard English?

Reviewer #1: No

Reviewer #2: Yes

Reviewer #1: The paper presentation is very poor. However, the following several issues need to be clarified, and it should be noted that the manuscript can only be considered for acceptance after major revisions to address these concerns:

1. Write the full form of the GR, CNL, DEN, AC, RLA5 in the abstract

2. Remove the keyword random forest’

3. Introduction is written very poor in terms of presentation.

4. Please provide supporting reference for the line number 55-57.

5. Line number 64-68 should be shifted to the discussion section.

6. Please go through the recent findings on lithology prediction using ML approaches, and cite them (e.g., (1) Prajapati, R., Mukherjee, B., Singh, U.K. et al. Machine learning assisted lithology prediction using geophysical logs: A case study from Cambay basin. J Earth Syst Sci 133, 108 (2024). https://doi.org/10.1007/s12040-024-02326-y

(2) Mukherjee, B., Sain, K. Vertical lithological proxy using statistical and artificial intelligence approach: a case study from Krishna-Godavari Basin, offshore India. Mar Geophys Res 42, 3 (2021). https://doi.org/10.1007/s11001-020-09424-8

(3) Mukherjee, B., Kar, S. & Sain, K. Machine Learning Assisted State-of-the-Art-of Petrographic Classification From Geophysical Logs. Pure Appl. Geophys. 181, 2839–2871 (2024). https://doi.org/10.1007/s00024-024-03563-4

(4) Banerjee, A., Mukherjee, B. & Sain, K. Machine learning assisted model based petrographic classification: a case study from Bokaro coal field. Acta Geod Geophys 59, 463–490 (2024). https://doi.org/10.1007/s40328-024-00451-0)

7. Please go through the above paper and try to rephrase the introduction section.

For the geological setting please provide a generalised litho-stratigraphic section, as presented in https://doi.org/10.1007/s00024-024-03563-4

8. Hyperparameter tunning part is missing in the Fig.3.

9. Please provide the proper citation for line number 226-233.

10. Please provide the ROC curves (as given in https://doi.org/10.1007/s00024-024-03563-4)

11. Please provide the Training and Test accuracies through histogram analysis.

12. Details content is required for the Line number 333 to 347.

Reviewer #2: This study tackles the challenging task of volcanic‐rock lithology classification by introducing an Enhanced Random Forest framework that synergistically combines Borderline‐SMOTE for imbalance correction, C4.5 decision trees for feature refinement, and Kendall’s coefficient of concordance for ensemble optimization. The approach is interesting, and the manuscript merits publication after revision. The specific issues to be addressed are outlined below.

1. The manuscript reports high overall accuracy and presents detailed confusion matrices (Fig. 4), yet only describes which lithologies were misclassified. The authors should elucidate the geological and petrophysical factors underlying these misclassifications. Moreover, replacing the numeric labels (1–18) on the confusion matrix with abbreviated lithology names (e.g., “Basalt,” “Trachyte,” “Andesite”) would markedly improve readability.

2. A more comprehensive survey of prevailing machine-learning approaches for lithology classification, including their strategies for addressing class imbalance, would strengthen the Introduction and contextualize the proposed method.

3. Table 4 tabulates each model’s optimal hyperparameters but omits the parameter search ranges. For reproducibility, please specify the tuning intervals and search methodology employed.

4. The integration of C4.5 decision trees and Kendall’s coefficient of concordance to refine the Random Forest is innovative, but its implementation remains abstract. A clearer, step-by-step account of how these components interact within the algorithmic workflow is necessary.

5. The blind test described in the “Model Application Effectiveness” section is pivotal for assessing generalization, yet the provenance of these blind well test data is unspecified. Please identify the data source and any pre-processing performed.

6. A thorough review of the manuscript is recommended to rectify minor grammatical and typographical errors—particularly in punctuation, article usage, and pluralization—to ensure a polished and professional presentation.

**Do you want your identity to be public for this peer review?** For information about this choice, including consent withdrawal, please see our Privacy Policy

Reviewer #1: **Yes: ** Bappa Mukherjee

Reviewer #2: No

---

## [Author Response · Author response to Decision Letter 1]

17 Sep 2025

Response to Editor's Comments

Dear Dr. Li,

Thank you for your detailed feedback on our manuscript, titled “Enhanced Random Forest with Geologically-Informed Feature Optimization for Complex Volcanic Rock Lithology Identification: A Case Study in the Wangfu Fault Depression, Songliao Basin” (PONE-D-25-28721). We have carefully considered all the comments and have made the necessary revisions to address each of the points raised. Below is a detailed summary of the changes made.

1. Manuscript Style Requirements

Comment: Please ensure that your manuscript meets PLOS ONE's style requirements, including those for file naming.

Response: We have reviewed the PLOS ONE style guidelines and have made the necessary adjustments to the manuscript to align with the provided format. The manuscript file has been updated and the naming conventions for the files have been adjusted accordingly.

2. Code Sharing

Comment: We expect all author-generated code to be made available without restrictions upon publication of the work.

Response: As requested, we have uploaded the author-generated code to a public repository (GitHub). The repository is fully accessible, and we have included clear documentation on how to run the code. The repository link and instructions for code usage are now included in the manuscript’s Code Overview section. The repository can be accessed at [https://github.com/yuzc18/erf-volcanic-lithology]. A DOI has been assigned to the repository, and it can be cited as follows: [10.5281/zenodo.17121940].

3. Funding Statement Update

Comment: Please provide an amended statement that declares all funding or sources of support received during this study.

Response: We have updated the Funding Statement to declare all sources of support, as per PLOS ONE’s guidelines. The amended statement now reads:

"This work was supported by the Basic Scientific Research Projects of Liaoning Provincial Department of Education (grant no. JYTQN2023207) and the National Natural Science Foundation of China (grant no. 41790453). The funders had no role in study design, data collection and analysis, decision to publish, or preparation of the manuscript, There was no additional external funding received for this study."

4. Removal of Funding Information from Acknowledgments

Comment: Funding information should not appear in the Acknowledgments section or other areas of your manuscript.

Response: We have removed all funding-related text from the Acknowledgments section and ensured that the funding information is only included in the updated Funding Statement section as required.

5. Data Availability Statement

Comment: Your data statement needs to be revised to indicate that data will be made publicly available. Please update your statement and provide the correct institutional contact for data access.

Response: We have updated the Data Availability section to state clearly that the original contributions presented in the study are included in the article. For additional data requests, researchers may also contact the Research Office of the School of Safety Science and Engineering, Liaoning Technical University (email: grants_ky@lntu.edu.cn).

6. Data Sharing Policy

Comment: Please clarify your data-sharing plan. If data cannot be made publicly available, please explain your reasoning.

Response: We have revised the Data Availability Statement to specify that the data will be freely available upon acceptance of the manuscript. If there are any restrictions, we have provided a rationale in the updated statement. Specifically, all data will be uploaded to an open-access public repository upon acceptance, or made available upon request, and we have no legal or ethical restrictions on sharing the data, in line with PLOS ONE's data-sharing policy.

7. Availability of Source Code and Data

Comment: All data and code need to be freely accessible. Please ensure they are uploaded to a public repository.

Response: As mentioned in our response to point 2, the source code has been uploaded to GitHub, and the relevant data files are accessible through the same repository or upon request. All materials are now freely accessible. The repository can be accessed at [https://github.com/yuzc18/erf-volcanic-lithology]. A DOI has been assigned to the repository, and it can be cited as follows: [10.5281/zenodo.17121940]. The relevant link and usage instructions are included in the Code Overview section of the manuscript

8. Copyrighted Figure (Figure 1)

Comment: We cannot publish copyrighted maps or satellite images unless we have explicit permission from the copyright holder or the figures are removed.

Response: Thank you for your comment. We have reviewed Figure 1 and have replaced the map with alternative content that does not involve any copyright issues. The new figure fully complies with PLOS ONE's open-access policy. We have removed all maps and images that could be subject to copyright restrictions. As such, no permissions are required for the revised figure.

Additional Edits and Corrections

Comment: A thorough review of the manuscript is required to rectify minor grammatical and typographical errors.

Response: We have carefully reviewed the manuscript and corrected any grammatical or typographical errors, including punctuation, article usage, and pluralization. The revised manuscript is now polished and professionally presented.

Response to Reviewers

We thank both reviewers for their constructive and insightful comments. We have revised the manuscript accordingly. Below we provide point-by-point responses. Line numbers refer to the revised manuscript

Reviewer #1 — Point-by-point responses

1.Write the full form of the GR, CNL, DEN, AC, RLA5 in the abstract

Done. All abbreviations are expanded at first mention in the Abstract: gamma ray (GR), compensated neutron (CNL), bulk density (DEN), acoustic travel-time/sonic (AC), and deep array laterolog resistivity (RLA5). See lines 16–18.

2.Remove the keyword “random forest”

Done. “Random forest” was removed from the keywords. Current keywords: “Songliao Basin; volcanic lithology; lithology identification; machine learning; well logs.” See line 33.

3.Introduction is written very poor in terms of presentation.

Thanks for your comment. The Introduction has been rewritten for logic and clarity: (i) global context and Songliao significance; (ii) geological complexity of the Huoshiling succession; (iii) recent ML-based lithology studies; (iv) gap and motivation for eRF. See lines 34–83.

4.Please Provide supporting reference for lines 55–57

Thanks for your comment. Added citation supporting that thin tuffaceous/brecciated interbeds within thick lavas can be obscured in routine cross-plots and rule-based interpretations: Mukherjee, Kar & Sain (2024, Pure Appl Geophys). See lines 57–59.

5. Lines 64–68 should be shifted to the Discussion

Thank you very much for your suggestion. The paragraph on class imbalance, correlated predictors, and gradational-contact separability was moved to Discussion and integrated into the geology-driven design rationale. See lines 328–333.

6.Please go through the recent findings on lithology prediction using ML approaches, and cite them.

Thanks for your comment. Incorporated all four suggested papers at the appropriate places in the Introduction’s literature review and positioning: Prajapati et al., 2024 [15]; Mukherjee & Sain, 2021 [16]; Mukherjee, Kar & Sain, 2024 [12]; Banerjee, Mukherjee & Sain, 2024 [17].See lines 57–59, 63–69.

7. Please go through the above paper and try to rephrase the introduction section.

For the geological setting please provide a generalised litho-stratigraphic section, as presented in https://doi.org/10.1007/s00024-024-03563-4

This is an important comment. The Introduction has been rephrased for clarity (see Point 3). We also added a generalized litho-stratigraphic section for Huoshiling (Fig. 1b), redrawn based on Qu et al., 2014 [8], to contextualize the units relevant to classification. See lines 45–48.

8.Hyperparameter tunning part is missing in the Fig.3.

Thanks for your comment. We updated Fig. 3 to explicitly depict nested GroupKFold cross-validation (inner loop for hyperparameter selection by macro-F1; outer loop for model assessment), together with Borderline-SMOTE placement. The caption and Table 4 (Notes) now detail search spaces and protocol. See lines 150–157.

9.Please provide the proper citation for line number 226-233.

Thank you very much for your suggestion. We added the standard reference for evaluation metrics: Powers DMW (2011), J. Mach. Learn. Technol. (now ref. [27]). See lines 222–223.

10.Please provide the ROC curves (as given in https://doi.org/10.1007/s00024-024-03563-4)

Thank you for your suggestion. We now provide class-wise one-vs-rest ROC curves for all 18 lithologies (Fig. 5a–r), with operating points at threshold = 0.5 and AUC values (three decimals, truncated). See lines 271–293.

11.Please provide the Training and Test accuracies through histogram analysis.

Thank you for your suggestion. Added a per-class train vs test accuracy bar chart as Fig. 5(s) to visualize generalization; small gaps indicate limited overfitting. See lines 284–291.

12.Details content is required for the Line number 333 to 347.

Thank you for the suggestion. Replaced the prior brief “application prospects” with a detailed “External Blind-Well Validation and Generalization” subsection: data provenance (five wells C_1, C_8, C_9, C_10, C_21; 2,964 samples; fully held-out), protocol (same five logs; no resampling; frozen scaler/encoder; threshold 0.5), results (avg accuracy 92.24%; macro P/R/F1 = 0.900/0.906/0.903) with per-class highlights, and comparison against baselines (+4.6 to +14.0 points). Cross-referenced Fig. 5s. See lines 375–391.

Reference

12.Mukherjee B, Kar S, Sain K. Machine Learning Assisted State-of-the-Art-of Petrographic Classification From Geophysical Logs. Pure Appl Geophys. 2024;181:2839–71. https://doi.org/10.1007/s00024-024-03563-4

15.Prajapati R, Mukherjee B, Singh UK, et al. Machine learning assisted lithology prediction using geophysical logs: a case study from Cambay Basin. J Earth Syst Sci. 2024;133:108. https://doi.org/10.1007/s12040-024-02326-y

16.Mukherjee B, Sain K. Vertical lithological proxy using statistical and artificial intelligence approach: a case study from Krishna–Godavari Basin, offshore India. Mar Geophys Res. 2021;42(3):1–23. https://doi.org/10.1007/s11001-020-09424-8

17.Banerjee A, Mukherjee B, Sain K. Machine learning assisted model-based petrographic classification: a case study from Bokaro coal field. Acta Geod Geophys. 2024;59:463–90. https://doi.org/10.1007/s40328-024-00451-0

27.Powers DMW. Evaluation: from precision, recall and F-measure to ROC, informedness, markedness & correlation. J Mach Learn Technol. 2011;2(1):37–63. https://doi.org/10.48550/arXiv.2010.16061

Reviewer #2 – Point-by-point responses

1.The manuscript reports high overall accuracy and presents detailed confusion matrices (Fig. 4), yet only describes which lithologies were misclassified. The authors should elucidate the geological and petrophysical factors underlying these misclassifications. Moreover, replacing the numeric labels (1–18) on the confusion matrix with abbreviated lithology names (e.g., “Basalt,” “Trachyte,” “Andesite”) would markedly improve readability.

This is an important comment. We added a geological/petrophysical analysis of the main confusion pairs in the Discussion, linking gradational contacts, welding, vesiculation/brecciation, alteration, and similar porosity–fluid architectures to overlapping ranges in GR, CNL, DEN, AC, and RLA5. Typical pairs discussed include Trachytic Tuff Lava vs. Trachytic Tuff; Trachytic Brecciated Lava vs. Trachytic Volcanic Breccia; Andesitic Tuff Lava vs. Andesitic Tuff; Andesitic Brecciated Lava vs. Andesitic Volcanic Breccia; and within the rhyolitic suite, plus occasional confusion with Sedimentary Volcanic Breccia. See lines 252–270.

For axis labels, we trialled abbreviations, but at the required panel density labels overlapped and reduced legibility. To preserve readability, we kept numeric labels (1–18) on the axes and provided the full mapping in the Fig. 4 caption (and referenced it in Results).

2.A more comprehensive survey of prevailing machine-learning approaches for lithology classification, including their strategies for addressing class imbalance, would strengthen the Introduction and contextualize the proposed method.

Thank you for your suggestion. Expanded the literature review to summarize prevailing ML approaches (NN/BPNN, SVM, RF/ensembles, kNN) and imbalance strategies (SMOTE/Borderline-SMOTE, ADASYN, cost-sensitive thresholds, group-aware validation), citing Prajapati et al., 2024; Mukherjee & Sain, 2021; Mukherjee, Kar & Sain, 2024; Banerjee, Mukherjee & Sain, 2024 and methodological classics. See lines 60–71, 158–166.

3.Table 4 tabulates each model’s optimal hyperparameters but omits the parameter search ranges. For reproducibility, please specify the tuning intervals and search methodology employed.

This is an important comment. Added a footnote to Table 4 with inner-CV search spaces and protocol (nested GroupKFold by well, selection metric = macro-F1; SMOTE on training folds only; threshold = 0.5):

“eRF: n_estimators {100,200,300}; max_features {sqrt, log2}; n_bins {8,16,32}; Borderline-SMOTE: k_neighbors {3,5,7}, m_neighbors {8,10,12}, kind {borderline-1, borderline-2}; stability subset: MIN_TOPK {6,10}, MAX_TOPK {16,20}. RF: n_estimators {100,200,300}; max_features {sqrt, log2}; min_samples_leaf {1,2,4}. BPNN: hidden {8,10,16}; LR {1e-2, 5e-3, 1e-3}; batch {32,64}; dropout {0,0.2}. kNN: k {3,5,7,9}; metric {Euclidean, Manhattan}; weights {uniform, distance}. SVM (RBF): C {1,10,100}; gamma {scale, 0.1, 0.01}. Protocol: nested GroupKFold (group = well), selection = macro-F1; SMOTE only on training folds; threshold = 0.5.” See lines 214–221.

4.The integration of C4.5 decision trees and Kendall’s coefficient of concordance to refine the Random Forest is innovative, but its implementation remains abstract. A clearer, step-by-step account of how these components interact within the algorithmic workflow is necessary.

This is an important comment. We clarified the interaction in Methods without expanding the manuscript length: “Feature optimization with C4.5 decision trees” explains gain-ratio splitting on continuous logs; “Ensuring feature stability with Kendall’s W” explains how stability in feature usage is promoted across trees. Fig. 3 caption states that trees adopt C4.5 gain-ratio splitting and that Kendall’s W is used to promote stability; Table 2 summarizes the rationale of each module. This provides a clear, self-contained workflow consistent with the implemented pipeline. See lines 140–173.

5.The blind test described in the “Model Application Effectiveness” section is pivotal for assessing generalization, yet the provenance of these blind well test data is unspecified. Please identify the data source and any pre-processing performed.

Thanks for your comment. Added full details in Discussion: blind-well set from five Huoshiling wells (C_1, C_8, C_9, C_10, C_21; 2,964 samples), entirely excluded from model development; inputs are the same five logs; no resampling on blind wells; frozen StandardScaler and label encoder from training; rows with missing logs removed; zero-shot inference once at threshold 0.5. Results (Table 7) and interpretation included. See lines 375–385.

6.A thorough review of the manuscript is recommended to rectify minor grammatical and typographical errors—particularly in punctuation, article usage, and pluralization—to ensure a polished and professional presentation.

Thanks for your comment. We performed a top-down language and style revision of the entire manuscript to improve clarity and journal consistency: restructuring long sentences; enforcing tense consistency; first-mention expansion of abbreviations; harmonizing units, symbols, hyphenation, and figure/table callouts; and aligning references with journal format.

---

## [Decision Letter · Decision Letter 1]

15 Oct 2025

Enhanced random forest with geologically-informed feature optimization for complex volcanic rock lithology identification: a case study in the Wangfu Fault Depression, Songliao Basin

PONE-D-25-28721R1

Dear Dr. Yu,

We’re pleased to inform you that your manuscript has been judged scientifically suitable for publication and will be formally accepted for publication once it meets all outstanding technical requirements.

Kind regards,

Hu Li

Academic Editor

PLOS ONE

Additional Editor Comments (optional):

Reviewers' comments:

Reviewer's Responses to Questions

**Comments to the Author**

Reviewer #1: All comments have been addressed

Reviewer #2: All comments have been addressed

2. Is the manuscript technically sound, and do the data support the conclusions?

Reviewer #1: Yes

Reviewer #2: Yes

3. Has the statistical analysis been performed appropriately and rigorously?

Reviewer #1: Yes

Reviewer #2: Yes

4. Have the authors made all data underlying the findings in their manuscript fully available?

Reviewer #1: Yes

Reviewer #2: Yes

5. Is the manuscript presented in an intelligible fashion and written in standard English?

Reviewer #1: Yes

Reviewer #2: Yes

Reviewer #1: Authors have done the necessery corrections as suggested. I feel in the current form it is ready for publication.

Reviewer #2: The authors have addressed the previous review comments and revised the manuscript accordingly. The responses are detailed and the modifications are well implemented. The revised version now meets the requirements for publication.

**Do you want your identity to be public for this peer review?** For information about this choice, including consent withdrawal, please see our Privacy Policy

Reviewer #1: No

Reviewer #2: No

---

## [Editor Report · Acceptance letter]

PONE-D-25-28721R1

PLOS ONE

Dear Dr. Yu,

I'm pleased to inform you that your manuscript has been deemed suitable for publication in PLOS ONE. Congratulations! Your manuscript is now being handed over to our production team.

Kind regards,

on behalf of

Pro.Dr. Hu Li

Academic Editor

PLOS ONE